# Reaction Time and Visual Memory in Connection to Alcohol Use in Persons with Bipolar Disorder

**DOI:** 10.3390/brainsci11091154

**Published:** 2021-08-30

**Authors:** Atiqul Haq Mazumder, Jennifer Barnett, Erkki Tapio Isometsä, Nina Lindberg, Minna Torniainen-Holm, Markku Lähteenvuo, Kaisla Lahdensuo, Martta Kerkelä, Ari Ahola-Olli, Jarmo Hietala, Olli Kampman, Tuula Kieseppä, Tuomas Jukuri, Katja Häkkinen, Erik Cederlöf, Willehard Haaki, Risto Kajanne, Asko Wegelius, Teemu Männynsalo, Jussi Niemi-Pynttäri, Kimmo Suokas, Jouko Lönnqvist, Jari Tiihonen, Tiina Paunio, Seppo Juhani Vainio, Aarno Palotie, Solja Niemelä, Jaana Suvisaari, Juha Veijola

**Affiliations:** 1Department of Psychiatry, Research Unit of Clinical Neuroscience, University of Oulu, 90014 Oulu, Finland; Martta.Kerkela@oulu.fi (M.K.); Tuomas.Jukuri@oulu.fi (T.J.); juha.veijola@oulu.fi (J.V.); 2Cambridge Cognition, University of Cambridge, Cambridge CB25 9TU, UK; jhb32@cam.ac.uk; 3Department of Psychiatry, University Hospital and University of Helsinki, 00029 Helsinki, Finland; erkki.isometsa@helsinki.fi (E.T.I.); nina.lindberg@helsinki.fi (N.L.); tuula.kieseppa@helsinki.fi (T.K.); asko.wegelius@fimnet.fi (A.W.); tiina.paunio@helsinki.fi (T.P.); 4Department of Psychiatry, University of Helsinki, 00014 Helsinki, Finland; jouko.lonnqvist@thl.fi; 5Mental Health Unit, Finnish Institute for Health and Welfare (THL), 00271 Helsinki, Finland; minna.torniainen-holm@thl.fi (M.T.-H.); erik.cederlof@thl.fi (E.C.); jaana.suvisaari@thl.fi (J.S.); 6Department of Forensic Psychiatry, Niuvanniemi Hospital, University of Eastern Finland, 70240 Kuopio, Finland; Markku.Lahteenvuo@niuva.fi (M.L.); Katja.Hakkinen@niuva.fi (K.H.); jari.tiihonen@ki.se (J.T.); 7Institute for Molecular Medicine Finland (FIMM), University of Helsinki, 00014 Helsinki, Finland; kaisla.lahdensuo@icloud.com (K.L.); ari.ahola-olli@helsinki.fi (A.A.-O.); hawker@utu.fi (W.H.); risto.kajanne@helsinki.fi (R.K.); teemu.mannynsalo@hel.fi (T.M.); jussi.niemi-pynttari@hel.fi (J.N.-P.); kimmo.suokas@tuni.fi (K.S.); aarno.palotie@helsinki.fi (A.P.); 8Mehiläinen, Pohjoinen Hesperiankatu 17 C, 00260 Helsinki, Finland; 9Department of Psychiatry, University of Turku, 20014 Turku, Finland; jarmo.Hietala@tyks.fi (J.H.); solnie@utu.fi (S.N.); 10Department of Psychiatry, Turku University Hospital, 20521 Turku, Finland; 11Faculty of Medicine and Health Technology, Tampere University, 33014 Tampere, Finland; olli.kampman@tuni.fi; 12Department of Psychiatry, Pirkanmaa Hospital District, 33521 Tampere, Finland; 13Social Services and Health Care Sector, City of Helsinki, 00099 Helsinki, Finland; 14Department of Clinical Neuroscience, Karolinska Institute, 17177 Stockholm, Sweden; 15Center for Psychiatry Research, Stockholm City Council, 11364 Stockholm, Sweden; 16Infotech Oulu, University of Oulu, 90014 Oulu, Finland; seppo.vainio@oulu.fi; 17Northern Finland Biobank Borealis, Oulu University Hospital, 90220 Oulu, Finland; 18Faculty of Biochemistry and Molecular Medicine, University of Oulu, 90014 Oulu, Finland; 19Kvantum Institute, University of Oulu, 90014 Oulu, Finland; 20Stanley Center for Psychiatric Research, The Broad Institute of MIT (Massachusetts Institute of Technology) and Harvard, Cambridge, MA 02142, USA; 21Analytical and Translational Genetics Unit, Massachusetts General Hospital, Boston, MA 02114, USA; 22Department of Psychiatry, Oulu University Hospital, 90220 Oulu, Finland

**Keywords:** cognition, visual memory, reaction time, alcohol, bipolar disorder

## Abstract

The purpose of this study was to explore the association of cognition with hazardous drinking and alcohol-related disorder in persons with bipolar disorder (BD). The study population included 1268 persons from Finland with bipolar disorder. Alcohol use was assessed through hazardous drinking and alcohol-related disorder including alcohol use disorder (AUD). Hazardous drinking was screened with the Alcohol Use Disorders Identification Test for Consumption (AUDIT-C) screening tool. Alcohol-related disorder diagnoses were obtained from the national registrar data. Participants performed two computerized tasks from the Cambridge Automated Neuropsychological Test Battery (CANTAB) on A tablet computer: the 5-choice serial reaction time task, or reaction time (RT) test and the Paired Associative Learning (PAL) test. Depressive symptoms were assessed with the Mental Health Inventory with five items (MHI-5). However, no assessment of current manic symptoms was available. Association between RT-test and alcohol use was analyzed with log-linear regression, and eβ with 95% confidence intervals (CI) are reported. PAL first trial memory score was analyzed with linear regression, and β with 95% CI are reported. PAL total errors adjusted was analyzed with logistic regression and odds ratios (OR) with 95% CI are reported. After adjustment of age, education, housing status and depression, hazardous drinking was associated with lower median and less variable RT in females while AUD was associated with a poorer PAL test performance in terms of the total errors adjusted scores in females. Our findings of positive associations between alcohol use and cognition in persons with bipolar disorder are difficult to explain because of the methodological flaw of not being able to separately assess only participants in euthymic phase.

## 1. Introduction

Persons suffering from bipolar disorder (BD) manifest cognitive impairment invariably during manic or depressive phase, and less intensely during euthymic phase [1,2,3]. BD patients show marked impairment on verbal and non-verbal learning and memory, attention, and executive functioning [4,5,6,7]. BD II patients with hypomanic episodes show similar [8,9] or, slightly less severe [10] cognitive deficits compared to BD I patients with manic episodes. In population based, representative, longitudinal samples, better cognitive functioning had been associated with increased risk of BD I [11,12]. However, a more recent study found no association between premorbid intelligence quotient (IQ) and risk of bipolar disorder [13].

About half of BD patients have lifetime diagnoses of alcohol use disorder (AUD) [14,15]. Alcohol use is the most prevalent (42%) substance use disorder among BD patients [16]. Alcohol misuse in BD is associated with worse outcome [17]. Even a lower volume of alcohol consumption is associated with illness severity in both male and female BD patients [18]. Mood symptoms in BD are primarily outcomes of AUD [19]. Also, more severe BD may be a risk factor for alcohol and other substance related disorders, a point that might have an impact on cognition [20].

BD patients with AUD show impaired verbal learning and memory [21,22], higher delay discounting [23], significant memory deficits more specifically the recognition of previously presented information [24] and, more deficits in executive functioning [25,26]. One study found significant impairments in executive control, working memory, attention and cognitive flexibility in comorbid AUD and BD patients, compared to healthy individuals, and patients with only AUD or BD patients [27]. On the other hand, BD patients without AUD also show impaired verbal learning and memory [21,22]. However, one study found no association between cognitive dysfunction and lifetime comorbid alcohol use disorder in BD patients [28]. A systemic literature review of eight studies (1998–2013) found association between current or past history of comorbid AUD and more severe cognitive impairment in BD patients [29].

Findings from normal population studies mostly suggest mild to moderate alcohol use not to be associated with cognitive impairment [30,31,32,33]. In persons with bipolar disorder, association of cognition with different drinking patterns other than alcohol use disorder, is yet to be studied.

The main aim of the present study was to explore the association between reaction time and visual memory with two drinking patterns in the same population diagnosed with bipolar disorder. The specific research aims were to study the following:(1)The association between hazardous drinking and reaction time and visual memory in persons with bipolar disorder.(2)The association between alcohol use disorder and reaction time and visual memory in persons with bipolar disorder.

## 2. Materials and Methods

### 2.1. Participants

The participants of this study were part of the 10,417-study population of the Suomalainen psykoosisairauksien perinnöllisyysmekanismien tutkimus study (“Finnish Study of the Hereditary Mechanisms behind Psychotic Illnesses”—SUPER), which was part of the international Stanley Global Neuropsychiatric Genomics Initiative. SUPER collected data from five university hospital districts in Finland during the period 2016–2019 from people with the lifetime diagnosis of psychotic illnesses, as classified by ICD-10 diagnostic codes F20–F29 (F20 Schizophrenia, F21 Schizotypal disorder, F22 Persistent delusional disorders, F23 Acute and transient psychotic disorders, F24 Induced delusional disorder, F25 Schizoaffective disorders, F28 Other nonorganic psychotic disorders, F29 Unspecified nonorganic psychosis), F30.2 (Mania with psychotic symptoms), F31.2 (Bipolar affective disorder, current episode manic with psychotic symptoms), F31.5 (Bipolar affective disorder, current episode severe depression with psychotic symptoms), F32.3 (Severe depressive episode with psychotic symptoms), and F33.3 (Recurrent depressive disorder, current episode severe with psychotic symptoms), to identify gene loci and gene variations predisposing patients to psychotic illnesses and comorbid diseases. These codes were used to identify subjects from the Care Register for Health Care (CRHC) and in clinical settings.

In clinical settings, such as healthcare centers, nursing homes, and psychiatric treatment facilities, staffs were asked to select patients with these diagnoses to be voluntarily recruited into the SUPER study. Subjects were also recruited via advertisements in local newspapers. Underage patients and patients unable to give informed consent as evaluated by the trained research personnel or attending physician, were excluded from the study.

Out of the original 10,417 study population, we included 1597 with a lifetime diagnosis of bipolar disorder and excluded those with a lifetime diagnosis of schizophrenia and schizoaffective disorder. Among the included participants, 66 had missing information on alcohol use, education, or depression. Of the remaining 1531 participants, 1361 were living independently after excluding hospitalized patients and those living in supported facilities. Finally, 1259 participants remained after excluding those aged 70 years or more. Among them 1163 participants (423 males, 740 females) completed reaction time (RT) test and 1048 participants (372 males, 676 females) completed paired association learning (PAL) test (Figure 1).

### 2.2. Bipolar Disorder Diagnoses

The diagnosis of bipolar disorder was obtained from the Care Register for Health Care (CRHC) of the National Institute for Health and Welfare of Finland. In Finland the ICD-system has been used in psychiatric diagnoses. In this study bipolar disorder diagnoses included both mania and bipolar disorder corresponding to the codes 296.1–296.8, 298.10 according to ICD-8; 296.2–296.4, 296.7A according to ICD-9 and F30, F31 according to ICD-10. In Finland ICD-8 was used during 1968–1986, ICD-9 during 1987–1995 and ICD-10 since 1996. During the use of ICD-9 in Finland, DSM-3 R criteria for bipolar disorder and other psychiatric disorders were used.

While selecting the original SUPER-study population, those who were not able to sign the written informed consent themselves had been excluded. In the present study, those who were hospitalized had been excluded as well by including only participants living independently. Thus, the most severe bipolar patients with severe depressive, or manic episodes were presumed to be excluded. However, cognitive impairment due to bipolar disorder in general population is not limited to the acute hospitalized episodes where patients might be unable to give an informed consent.

### 2.3. Hazardous Drinking Screening

About 90% people drinking excessive alcohol could positively be screened as hazardous drinkers or binge (heavy episodic) drinkers instead of fulfilling the diagnostic criteria for AUD [34]. Hazardous drinking is a pattern of alcohol consumption that increases the risk of harmful consequences for the user or others. Hazardous drinking patterns are of public health significance despite the absence of any current disorder in the individual alcohol user [35,36,37,38].

Hazardous drinking was screened using the AUDIT-C questionnaire to assess an individual’s alcohol consumption frequency (“How often do you have a drink containing alcohol?”), quantity (“How many drinks containing alcohol do you have on a typical day when you are drinking?”), and bingeing (“How often do you have six or more drinks on one occasion?”). AUDIT-C is derived from the hazardous alcohol use domain of the Alcohol Use Disorders Identification Test (AUDIT) questionnaire [39]. It has three questions and is scored on a scale of 0 to 12. Each AUDIT-C question has five answer choices valued from zero to four points.

Cutoff scores for hazardous drinking vary considerably [40,41]. In the present study, we used the cutoff scores recommended by Finnish National Guidelines: a score of 6 or more in males and 5 or more in females [42].

### 2.4. Alcohol Use Disorder Diagnoses

Alcohol use disorders include F10.1 (harmful alcohol use) and F10.2 (alcohol dependence), hence we have used the term “alcohol-related disorder” in our text elsewhere for better understanding. The diagnoses of alcohol use disorder were obtained from the CRHC data according to codes ICD-8 291 and 303; ICD-9 291, 3030, and 3050A; and ICD-10 F10 for the period from 1969 to 2018.

### 2.5. Cognitive Measures

Processing speed and visual learning are the two cognitive domains affected invariably in psychotic illnesses; hence, we selected the Five-Choice Serial Reaction Time Task (5-CRTT) and the Paired Associative Learning (PAL) task from the Cambridge Neuropsychological Test Automated Battery (CANTAB) for the assessment of reaction time (RT) and visual memory, respectively [43].

These tasks were chosen to produce relevant information on cognition in psychotic disorders in the very restricted assessment schedule. The instructions for both tests were translated into Finnish. The CANTAB tests were performed before venipuncture in order to avoid malfunction of the arm due to pain or bandaging. The study nurses were given standardized instructions on how to guide the study subjects in performing the CANTAB test beforehand.

In the RT test, we used two continuous measurements: the median of the five-choice reaction time and the standard deviation (SD) of the five-choice reaction time. The median of the five-choice reaction time is the median duration between the onset of the stimulus and the release of the button. The standard deviation of the five-choice reaction time is the standard deviation of the time taken to touch the stimulus after the button has been released. Both variables were calculated for correct, assessed trials where the stimulus could appear in any of the five locations.

In the PAL-test we assessed visual memory using the primary outcome variables of ‘total errors adjusted’ and first trial memory score. First Trial Memory Score (FTMS) is how many patterns the participant correctly places on the first attempt at each problem, while Total Errors (Adjusted) (TEA) reflects how quickly the participant learns when the participant has multiple attempts at each problem. For PAL TEA we assessed a dichotomized variable because the distribution of the PAL TEA does not follow any known distribution with multiple peaks, using data from Northern Finland Birth Cohort 1966 (NFBC 1966) as a reference data. The NFBC 1966 consists of all born with expected date in the year 1966. The data used in this study consist of a 46-year follow-up when cohort members took the PAL-test during clinical examination (N = 5608). Scores for total errors adjusted of NFBC66, the 50th percentile (10 error score or less) was used as a cut-off for good performance in PAL test in the recent study, meaning the SUPER study population made better error score than a 50% of NFBC 1966 study population. The PAL FTMS variable was used as a continuous variable.

### 2.6. Confounding Factors

Age, education [44], housing status [45] and depressive symptoms [3] have effects on cognition. Hence, we considered them to be the confounding variables in this study.

#### 2.6.1. Age

Cognition is negatively associated with increased age in healthy populations [46] and debatably in alcohol users [47]. The age of the participants was calculated using the participation date and year of birth of the participant. Age was used as continuous variable.

Both illness duration of bipolar disorder [48] and late-onset bipolar disorder [49] is associated with more severe cognitive impairments hence we could also use age of first bipolar episode as a cofounder. However, we did not take age of onset as a confounder considering the complex effects of age. Younger age is associated with both better performance in cognitive tests and with more alcohol use. In addition, younger age is associated with younger age at onset. If we included age of onset in the analysis, we in a way would include the age twice. What it meant if age is correlated both with the outcome of interest and our main explanatory variable (alcohol use). The multicollinearity might exist in the adjusted analysis. However, multicollinearity affects only the specific independent variables that are correlated. Therefore, if multicollinearity was not present for the independent variables that we were particularly interested in, we might not need to resolve it. If we would desire to keep both (age and age of first bipolar episode) in the analysis, the multicollinearity would be hard to avoid.

#### 2.6.2. Education

Education is strongly associated with cognitive performance [50]. The question and possible answers addressing the education of the participants were as follows: “What is your basic education?” (1 = less than primary school, 2 = matriculation examination, 3 = middle school, 4 = partial general upper secondary school or general upper secondary education certificate, 5 = partial middle school or primary school less than nine years, 6 = primary school, 7 = four-year elementary school). During the analysis, we combined classes 1, 3, 4, 5, 6, and 7 as “No matriculation examination” versus class 2 (“Matriculation examination”).

It would be more informative if we could categorize education into three groups. However, it might be difficult for a general reader to understand the diverse categories in the Finnish education system reflecting changes over the past seventy years plus additional categories reflecting the small proportion of immigrants who might have lower general education than that provided in the Finnish education system. We used the general education variable because the youngest participants could still be students.

#### 2.6.3. Household Pattern

Household patterns, especially living without a spouse, might affect cognition [45,51], and thus we considered household patterns as a confounder.

#### 2.6.4. Depression

Depressive symptoms might be associated with poorer cognitive performance [3], hence we considered depression as a confounder. We used the five-item Mental Health Inventory-5 (MHI-5) to detect depressive symptoms. In the analysis MHI-5 was dichotomized. We used ≤72 cutoff score for depression which was also used in a recent population-based study in Finland [52].

### 2.7. Statistical Methods

We evaluated the association between cognition and alcohol use by using four different cognition variables: median and standard deviation of RT, PAL FTMS and PAL TEA. Alcohol use was measured by different variables; dichotomous hazard drinking variable derived from AUDIT-C score; and dichotomous variable indicating if the study subject had had alcohol-related ICD-diagnosis. We assessed crude models and adjusted models with age, household pattern and education. Association between RT-test and alcohol use was analyzed with log-linear regression, and eβ with 95% confidence intervals (CI) are reported. Association between PAL FTMS-test was analyzed with linear regression, and β with 95% CI are reported. All continuous variables used in models were normalized using z-score. Association between PAL-TEA and alcohol use was analyzed with logistic regression and odds ratios (OR) with 95% CI are reported. All the analyses were conducted separately for males and females. Males and females showed differences in performing the selected cognitive tests [52,53,54]; additionally, males and females showed differences in alcohol use patterns [55,56,57,58].

## 3. Results

### 3.1. Background Factors and Alcohol Use Patterns in Persons with Bipolar Disorder

Of the participants about two-fifth were males and three-fifth were females. Mean age was 45 years for males and 44 years for females. One-third of the males and about half of the females had the highest basic educational of 12 years (matriculation). One-third of the males and two-third of the females were living with their spouses. Most of the participants were on psychotropic medication. Three-fourth males and females were detected screening-positive depression (Table 1).

Hazardous drinking and AUD were more common in males than in females. Hazardous drinking was significantly more common in depressed (MHI-5 < 70) females than in non-depressed (MHI-5 < 70) females (28.8% vs. 19.5%, *p* = 0.011). There was no significant difference in the prevalence of AUD between depressed and non-depresses females. In males these comparisons were statistically non-significant. About two-fifth of the males and one-fourth of the females were screened positive for hazardous drinking. Also, two-fifth of the males and one-fourth of the females had a lifetime diagnosis of alcohol-related disorder (Table 1).

Lower age was associated with hazardous drinking, both in males and females (see the Appendix A).

The median RT was 416 ms (SD = 45 ms); the PAL median FTMS was 11 and the median total errors adjusted was 17 (see the Appendix A).

The association between background factors and AUD with RT *p*-values has been reported in the Appendix A. The association between background factors and alcohol use patterns with PAL scores has been reported in the Appendix A. The Cohen’s d measure of effect size has been shown in the Appendix A.

### 3.2. Association of Reaction Time and Visual Memory with Hazardous Drinking in Bipolar Disorder

After adjustment of age, education, household patterns and depression, hazardous drinking was associated with lower median RT both in males (OR = 0.83; 95% CI, 0.69–1.00) and in females (OR = 0.82; 95% CI, 0.71–0.95) and with a less variable reaction time in females (OR = 0.85; 95% CI, 0.73–0.98) (Table 2). The association between hazardous drinking and RT scores has been reported in the Appendix A.

The association of reaction time and visual memory with hazardous drinking without adjusting depression has been shown in the Appendix A. The results were basically same without depression as a covariate.

### 3.3. Association of Reaction Time and Visual Memory with Alcohol-Related Disorder in Bipolar Disorder

Median RT or SD RT did not differ statistically significantly between participants with or without a lifetime history of alcohol use disorder, in males or in females (Table 3).

After adjustment of age, education, household patterns and depression, females with AUD performed more poorly in the PAL test than females without AUD (OR = 0.53; 95% CI, 0.34, 0.81) (Table 3). The association of reaction time and visual memory with Alcohol-Related Disorder without adjusting depression has been shown in the Appendix A. The results were basically same without depression as a covariate.

## 4. Discussion

### 4.1. Main Findings

Our findings did not support our assumption that problematic drinking might be associated with impaired cognitive function in a sample of outpatients with bipolar disorders below 70 years old. On the contrary, some positive associations were found between hazardous drinking and reaction time scores in males and females. As our study is new of its kind, it is difficult compare it to literature to compare with. However, our findings were partly in line with the findings of some general population studies suggesting that moderate drinking was not associated with cognitive impairment [59,60], not hazardous drinking and not AUD.

Another finding in our study, association of AUD with poor visual memory in females but not in males, was not fully aligned with findings from most of the previous studies [61,62].

### 4.2. Comparison with Other Studies

As per our knowledge, there are no other studies investigating the association between cognitive testing in terms of reaction time and visual memory, and different alcohol use patterns, namely hazardous drinking, and alcohol-related disorder, in persons with bipolar disorder, it was difficult to compare our findings with other studies. Most of the studies investigating cognitive impact of alcohol in persons with bipolar disorder with comorbid AUD revealed a correlation between alcohol use and cognitive impairment.

A recent study showed that verbal learning and memory, rather than selective attention and executive function, were impaired among BD patients with and without AUD [21]. Impairment of verbal learning and memory was also found in previous clinical studies of participants with comorbid BD and SUDs [22,26,63]. AUD patients without BD have also shown impaired verbal learning and memory [64].

BD patients with AUD show higher delay discounting [23] and significant memory deficits more specifically the recognition of previously presented information [24]. BD patients with a history of alcohol dependence showed decreased executive functioning [26]. BD patients with comorbid alcohol dependence also showed more severe impairment in executive functioning [25] and less recovery from cognitive deficits than only BD patients [63]. However, one recent research showed that BD patients with comorbid alcohol dependence had initial delay but subsequent recovery in executive domain [65].

One recent study revealed that patients with comorbid alcohol dependence and affective disorder manifested significant impairments in executive control, working memory, attention and cognitive flexibility compared to healthy individuals, and patients with only alcohol dependence or affective disorder [27].

BD patients with AUD exhibit less recovery from cognitive deficits than only BD patients [63]. However, one recent research showed that BD patients with comorbid AUD had initial delay but subsequent recovery in executive domain [65].

Marshall et al. found significantly worse performance on tasks of visual memory and reasoning in BD patients with comorbid AUD [66]. Chang et al. found widespread cognitive deficits, especially in terms attention/concentration and working memory, in BD patients with comorbid AUD [67]. Levy et al. found that BD patients with comorbid AUD had more deficits in verbal memory, visual memory, executive functioning, and a poorer neurocognitive recovery [63]. Shan et al. found more impaired visual memory, verbal memory, attention, psychomotor speed, working memory, and executive functioning in type-II BD patients with comorbid AUD [68]. Bonnín et al. found more deficits in verbal memory and executive functions in euthymic BD patients with and without past history of AUD, compared to healthy controls [69]. Levy et al. found an association of more severe mnemonic and executive dysfunction in BD patients with comorbid AUD [22]. van Gorp et al. found verbal memory deficits in BD patients with or without comorbid AUD, and an additional executive deficit in comorbid group [26].

Based on a post-hoc analysis, one study suggested lifetime comorbid AUD not to be associated with cognition in BD patients [28]. Another study found no association between cognitive dysfunction and mood disorder in AUD patients [70]. A third study suggested association of creativity with comorbid alcohol dependence in BD patients [71].

A recent study with similar settings among schizophrenia patients found positive association between hazardous drinking and lower median RT in females and less variable RT in males diagnosed with schizophrenia and schizoaffective disorders. The same study also found a positive association between AUD and was poorer PAL test performance in females diagnosed with schizophrenia [72].

General population studies on adolescent binge drinking yielded mixed results. Most cross-sectional studies suggested a negative association between adolescent binge drinking and cognitive functioning [73,74]. Some prospective studies suggested that binge drinking preceded cognitive impairment in young adults [75,76,77,78], while other prospective studies suggested that cognitive impairment preceded binge drinking in young adults [79,80,81]. A recent prospective study using a combination of observational and genetic approaches, found no evidence of binge drinking in between the ages of 16 and 23 and cognitive deficits at age 24 [82]. Prospective study in population study also found no evidence of heavy drinking in adolescent and cognitive impairment in later life [83].

A brief review of 29 studies (2003–2013) revealed that acute alcohol mostly impaired executive function in normal population [84]. In contrast, a systematic review of 143 studies (1977–2011) revealed that light to moderate alcohol use did not impair cognition in young male and female individuals and reduced the risk of all forms of dementia and cognitive decline in older individuals [85]. Another systematic review of 28 reviews (2000–2017) revealed that light to moderate alcohol use in middle to late adulthood was associated with a decreased risk of cognitive impairment and dementia [31]. Meta-analysis of 27 cohort studies (2007–2018) revealed that moderate alcohol uses improved cognition insignificantly among male and slightly among female compared to current non-drinkers [30]. Moderate alcohol has been found to be associated with reduced amyloid-beta deposition in human brain [86].

Low data quality of self-reported alcohol use is associated with illness severity in BD patients [87,88] hence more severe BD patients might misreport or under-report about their alcohol consumption. Selection bias can lead to compare healthy drinkers with unhealthy nondrinkers, according to ‘sick quitter’ hypothesis [89].

Study findings suggesting association between alcohol consumption and better performance in cognitive testing could be attributed by unmeasured or residual confounding factors [90,91] like smoking [92], drink type [93], drink pattern [94], personality [92,95], intelligence [96,97], educational attainment [98,99], potential abstainer errors [100,101,102,103], reverse causality bias [104], recall error [105], within person temporal variation [106,107]. Study findings suggesting negative association between alcohol and cognition could be attributed by poor motivation [108,109,110].

### 4.3. Strengths

We used a large dataset comprising persons with bipolar disorder to investigate cognitive impact of different alcohol use patterns. We studied two different alcohol use patterns in the same study population and used age and education as potential confounding variables.

We excluded people of 70 years and above to minimize reverse causality bias. However, we performed analyses keeping those over 70 years of age and found almost no differences. Similarly, we also analyzed our data keeping both age and age of onset as confounders but got almost same results.

We have included all persons with bipolar disorder living independently and excluded those whose living circumstances (living in supported housing, hospitals, or unknown residence) might affect their alcohol use. We have also confounded household patterns (those living with spouses versus those without). However, current housing situation is not that relevant while using alcohol-related disorder information during lifetime.

Our inclusion criterion of independent living excluded hospitalized patients, so patients with severe manic or depressive episode were not included. Another inclusion criterion was the ability to give written informed consent which also restricted inclusion of bipolar disorder patients with severe manic or depressive symptoms.

We performed sensitivity analyses including also those aged 70 years and above and, those not living with their spouses. We also performed Cohen’s d measure of effect size for our study findings.

### 4.4. Limitations

Our study was cross-sectional, not longitudinal. We did not adopt more comprehensive approach to measure working memory performance. We used only a reaction time test and a memory test from the CANTAB, while most of the literature we reviewed showed that AUD in BD patients was associated with executive function deficits. So, patients with AUD might be more impulsive, and less accurate. Furthermore, memory impairment is expected to occur only in patients with severe alcohol-related cognitive impairment (formerly known as alcohol dementia), a sub-group of patients with AUD.

We did not use information about the onset of alcohol use, any recent changes in drinking habits or any previous history of abstinence. We also did not differentiate previous alcohol users from never-alcohol users and did not exclude individuals who reduced drinking due to illness/doctor’s advice which might attribute the results through reverse causality bias [91,111]. We did not correct self-report bias [112,113] and misreports and longitudinal changes (MLC) which could affect the study results [114,115]. We did not confound household income, which, as indicative of socioeconomic status, could increase alcohol-related mortality and morbidity despite lower reportedly alcohol consumption (alcohol harm paradox) [116]. We confounded education, which is another strong indicator of socioeconomic status, in a dichotomous fashion, not in a stratified one.

We did not confound antipsychotic medication because almost all of the persons with bipolar disorder were on antipsychotic medication. We did not confound benzodiazepines use as it could impair cognitive performance because of its acute sedative effect. We also did not confound smoking or other substance use during lifetime, and we did not confound other F1-diagnoses. We did not incorporate mendelian randomization to minimize possible reverse causality bias. We did not use continuous variables for PAL test.

We categorized education as completed general secondary education with matriculation examination versus lower. It would be more informative if we could categorize education into three groups. However, it might be difficult for a general reader to understand the diverse categories in the Finnish education system reflecting changes over the past seventy years plus additional categories reflecting the small proportion of immigrants who might have lower general education than that provided in the Finnish education system. We used the general education variable because the youngest participants could still be students.

The phase of the illness in bipolar disorder is an important issue to be considered while measuring cognitive functions in bipolar disorder. But unfortunately, we were not able to use manic symptoms in our analysis, as the data did not include assessment of current manic symptoms.

We did not correct for multiple comparisons (Bonferroni correction). Since most of the confidence intervals did not come close to 1.00 it was obvious that most results would remain significant also when these corrections were applied. It might be worth pre-emptying non-significant comparisons.

### 4.5. What Is Already Known on This Subject?

Alcohol use disorder is associated with cognitive decline in persons with bipolar disorder.Mild alcohol use is not associated with impaired cognition in the normal population.

### 4.6. What Does This Study Add?

−Hazardous drinking was not associated with cognitive decline in persons with bipolar disorder in outpatients, below 70, when a reaction time test and a memory test were used without an assessment of executive functioning and without correcting for manic symptoms.

## 5. Conclusions

Hazardous drinking was not associated with a cognitive decline in persons with bipolar disorder in outpatients, below 70, adjusted with age, education, household pattern and depression, when a reaction time test and a memory test were used without an assessment of executive functioning and without correcting for manic symptoms. Although, some positive associations were found between hazardous drinking and cognition, most of them were lost when adjusting (Appendix A), and the positive association was only in line with moderate drinking in the general population, not hazardous drinking and not AUD.

Selection bias or severity of illness (other than duration) could have some undetected attributions on our results. Participants with more severe illness might drink less compared to those with risky drinking even though we have excluded those not living independently. Future studies should use larger observational samples, meta-analyses of related cognitive measures in GWAS, other proxy for severity of illness to increase power. Replication of this study by incorporating mendelian randomization with observational analysis might reduce possible bias from residual confounding and reverse causation [117]. Hence, our study might serve as a reference for future research.

## Figures and Tables

**Figure 1 brainsci-11-01154-f001:**
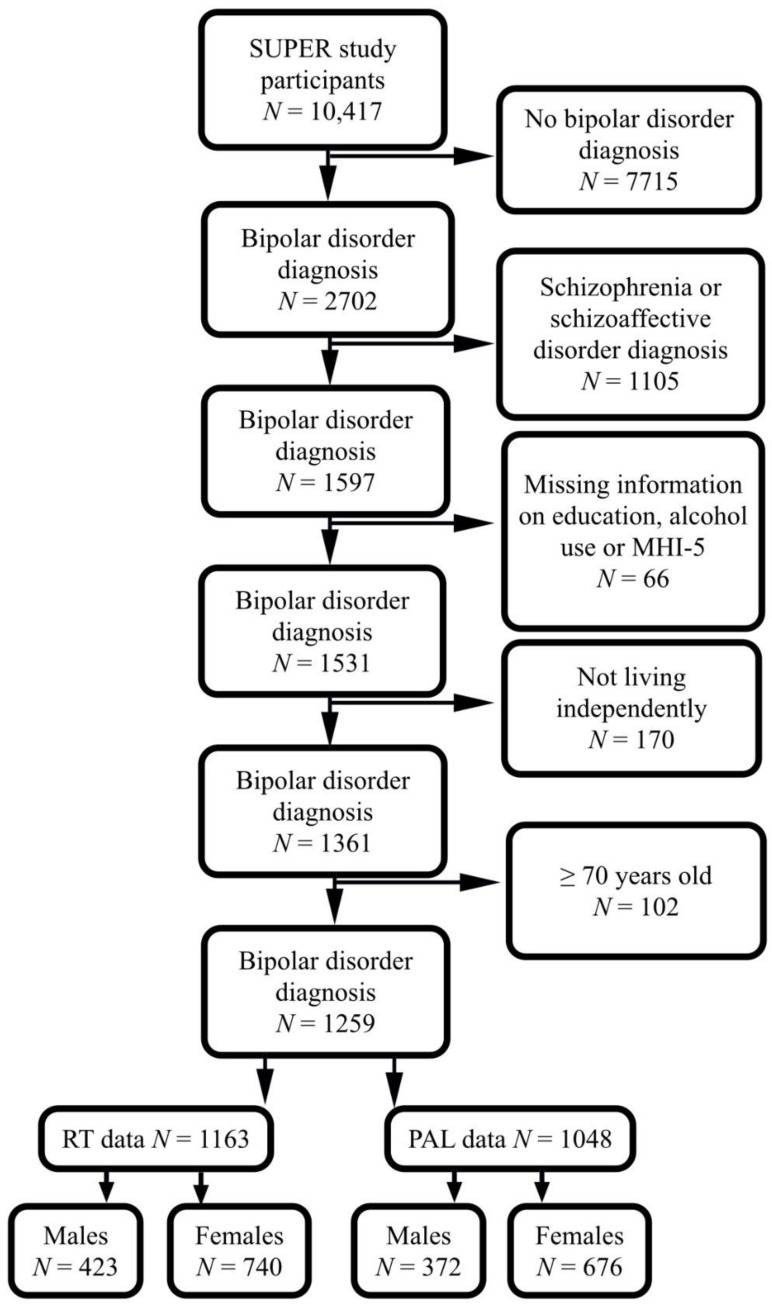
Flowchart showing the selected study population. SUPER, Suomalainen psykoosisairauksien perinnöllisyysmekanismien tutkimus; MHI-5, mental health inventory-5; RT, reaction time; PAL, paired associative learning.

**Table 1 brainsci-11-01154-t001:** Background factors and alcohol use patterns in persons with bipolar disorder.

	Male	Female
	N = 466	N = 793
Age (mean (SD))	45.35 (13.05)	44.37 (12.74)
Education		
No matriculation examination (%)	309 (66.3)	436 (55.0)
Matriculation examination (%)	157 (33.7)	357 (45.0)
Household pattern		
With spouse	172 (36.9)	341 (43.0)
Without spouse	294 (63.1)	452 (57.0)
Depression ^Ω^		
Depressed	331 (71.0)	583 (73.5)
Non-depressed	135 (29.0)	210 (26.5)
Current Psychotrophic medications		
No (%)	27 (5.8)	36 (4.5)
Yes (%)	439 (94.2)	756 (95.3)
Missing (%)	0 (0.0)	2 (0.2)
Hazardous drinking ^ψ^		
No (%)	287 (61.6)	584 (73.6)
Yes (%)	179 (38.4)	209 (26.4)
Lifetime alcohol-related disorder		
No (%)	288 (61.8)	597 (75.3)
Yes (%)	178 (38.2)	196 (24.7)

SD = Standard deviation. ^Ω^ MHI-5 cutoff score for depression was ≤72. ^ψ^ AUDIT-C cutoff scores for hazardous drinking were ≥6 for males and ≥5 for females.

**Table 2 brainsci-11-01154-t002:** Association of reaction time and visual memory with hazardous drinking ^ψ^ in bipolar disorder.

	Five Choice Reaction Time *							
	Median				SD			
	Crude		Adjusted ^a^		Crude		Adjusted ^a^	
	e^β^ (95% CI)	*p*-Value	e^β^ (95% CI)	*p*-Value	e^β^ (95% CI)	*p*-Value	e^β^ (95% CI)	*p*-Value
Male								
Hazardous drinking	0.76 (0.62, 0.92)	0.005	0.83 (0.69, 1.00)	0.047	0.79 (0.65–0.96)	0.021	0.88 (0.73–1.07)	0.203
Female								
Hazardous drinking	0.77 (0.67, 0.90)	0.001	0.82 (0.71, 0.95)	0.008	0.76 (0.66, 0.89)	<0.001	0.85 (0.73, 0.98)	0.027
	**Good performance in PAL**							
	**FTMS** ******				**TEA** *******			
	**Crude**		**Adjusted ^a^**		**Crude**		**Adjusted ^a^**	
	**β (95% CI)**	** *p* ** **-Value**	**β (95% CI)**	** *p* ** **-Value**	**OR (95% CI)**	** *p* ** **-Value**	**OR (95% CI)**	** *p* ** **-Value**
Male								
Hazardous drinking	0.05 (−0.17, 0.27)	0.643	−0.07 (−0.26, 0.12)	0.471	0.93 (0.59, 1.48)	0.775	0.68 (0.40, 1.14)	0.144
Female								
Hazardous drinking	0.12 (−0.04, 0.28)	0.153	0.01 (−0.14, 0.16)	0.918	1.17 (0.81, 1.67)	0.405	0.97 (0.66, 1.42)	0.882

RT = Reaction time PAL = Paired association learning SD = Standard deviation CI = Confidence interval. ^ψ^ AUDIT-C cutoff scores for hazardous drinking were ≥6 for males and ≥5 for females. ^a^ Adjusted with age, household pattern, depressive symptoms and education. * Analyzed with log-linear regression. ** Analyzed with linear regression. *** Analyzed with logistic regression.

**Table 3 brainsci-11-01154-t003:** Association of reaction time and visual memory with alcohol-related disorder in bipolar disorder.

	Five Choice Reaction Time *				Five Choice Reaction Time *			
	Median				SD			
	Crude		Adjusted ^a^		Crude		Adjusted ^a^	
	e^β^ (95% CI)	*p*-Value	e^β^ (95% CI)	*p*-Value	e^β^ (95% CI)	*p*-Value	e^β^ (95% CI)	*p*-Value
Male								
Alcohol-related disorder	1.18 (0.97, 1.44)	0.091	1.01 (0.83, 1.23)	0.917	1.13 (0.92, 1.38)	0.254	0.92 (0.76, 1.12)	0.395
Female								
Alcohol-related disorder	1.05 (0.90, 1.23)	0.496	1.03 (0.89, 1.20)	0.707	1.03 (0.88, 1.21)	0.680	1.02 (0.88, 1.18)	0.838
	**Good performance in PAL**							
	**FTMS** ******				**TEA** *******			
	**Crude**		**Adjusted ^a^**		**Crude**		**Adjusted ^a^**	
	**β (95% CI)**	** *p* ** **-Value**	**β (95% CI)**	** *p* ** **-Value**	**OR (95% CI)**	** *p* ** **-Value**	**OR (95% CI)**	** *p* ** **-Value**
Male								
Alcohol-related disorder	−0.27 (−0.50, −0.06)	0.012	−0.06 (−0.26,0.14)	0.558	0.57 (0.34, 0.93)	0.028	0.81 (0.46, 1.24)	0.473
Female								
Alcohol-related disorder	−0.15 (−0.32, 0.02)	0.085	−0.10 (−0.25, 0.06)	0.225	0.51 (0.34, 0.77)	0.002	0.53 (0.34, 0.81)	0.004

RT = Reaction time. PAL = Paired association learning. SD = Standard deviation. CI = Confidence interval. ^a^ Adjusted with age, household pattern, depressive symptoms and education. * Analyzed with log-linear regression. ** Analyzed with linear regression. *** Analyzed with logistic regression.

## Data Availability

The raw data and materials used for this study are available upon request.

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
