# Peer review of "Reaction Time and Visual Memory in Connection to Alcohol Use in Persons with Bipolar Disorder"

_brainsci, 2021, doi:10.3390/brainsci11091154_

Round 1
Reviewer 1 Report
The paper provides a comprehensive analysis of the association between hazardous drinking and cognitive decline in persons with
bipolar disorder, which has not yet been previously studied. Evidence of the absence of this association was obtained in this study. The authors of the manuscript described in detail the study limitations related to both the study design and the methodological approach, potential sampling and information biases. The results of this study need to be confirmed in future systematic reviews and meta-analyzes, as well as in individual studies conducted on similar samples of subjects with bipolar disorder.
Author Response
Author’s responses to the comments of REVIEWER 1:
English language and style
( ) Extensive editing of English language and style required
( ) Moderate English changes required
(x) English language and style are fine/minor spell check required
( ) I don't feel qualified to judge about the English language and style
Author’s response:
English language, style and spell check have been done accordingly in the revised version of our manuscript.
Yes Can be improved Must be improved Not applicable
Does the introduction provide sufficient background and include all relevant references?
(x) ( ) ( ) ( )
Is the research design appropriate?
(x) ( ) ( ) ( )
Are the methods adequately described?
(x) ( ) ( ) ( )
Are the results clearly presented?
(x) ( ) ( ) ( )
Are the conclusions supported by the results?
(x) ( ) ( ) ( )
Author’s response:
Introduction, research design, methods and results have further been modified. Abstract, introduction, materials and methods, results, conclusions, strengths, limitations, and references have further been modified in the revised version. We have also provided additional supplementary materials.
Comments and Suggestions for Authors
The paper provides a comprehensive analysis of the association between hazardous drinking and cognitive decline in persons with bipolar disorder, which has not yet been previously studied. Evidence of the absence of this association was obtained in this study. The authors of the manuscript described in detail the study limitations related to both the study design and the methodological approach, potential sampling, and information biases. The results of this study need to be confirmed in future systematic reviews and meta-analyzes, as well as in individual studies conducted on similar samples of subjects with bipolar disorder.
Author’s response:
Thank you for your inspiring comments and evaluation. We fully agree with you that the results of this study need to be confirmed in future systematic reviews and meta-analyzes, as well as in individual studies conducted on similar samples of subjects with bipolar disorder.
Date of submission: 15 June 2021
Date of review: 22 Jul 2021 16:35:25
Date of submitting author’s response: 13 Aug 2021

Reviewer 2 Report
The authors propose to analyse the association between the results on two neuropsychological tests (reaction time and paired-associated learning) with alcohol use in a large sample of bipolar patients (N~1200) recruited from a larger genetic study.
In their introduction, the authors expose in ONE sentence the complex issue of cognitive impairment in the course of bipolar disorder: "Cognitive impairment is a prominent [1] and generally stable [2] core symptom [3]
in persons with bipolar disorder (BD)".
Thus, they miss a central caveheat in assessing such neuropsychological impairment. GENERALLY STABLE means that cognitive impairment is persistent BETWEEN mood episodes. the teams that have properly measured such impairment are cautious enough to assess them in euthymic patients. for an exemple, see Roux et al. 2020 PMID 29734950. in that study, Roux et al. used cut-off scores of depression and mania to select euthymic patients for their study.
Of course, during manic and depressive episodes, the symptoms impair neuropsycological results, toward impulsive rapid responses during mania, and toward slow responses in depression.
Here the methods section does not provide any kind of subject selection based on euthymia checking, not even a post-hoc analysis of a correlation between the neuropsychological results and the symptoms levels.
As such, the design does not allow to draw any conclusion from the results.
the patients with higher alcohol use, or even more with an alcohol use disorders are known to have a worse course of their bipolar disorder (more episodes, more affectiveliability, more suicidality, less response rate to mood stabilizers). they are expected to be more symptomatic in such a lare cohort. they could display different results in RT and PAL only because they are less euthymic.
Author Response
Author’s responses to the comments of REVIEWER 2:
English language and style
( ) Extensive editing of English language and style required
( ) Moderate English changes required
(x) English language and style are fine/minor spell check required
( ) I don't feel qualified to judge about the English language and style
Author’s response:
English language, style and spell check done accordingly.
Yes Can be improved Must be improved Not applicable
Does the introduction provide sufficient background and include all relevant references?
( ) ( ) (x) ( )
Is the research design appropriate?
( ) ( ) (x) ( )
Are the methods adequately described?
( ) ( ) (x) ( )
Are the results clearly presented?
( ) ( ) (x) ( )
Are the conclusions supported by the results?
( ) ( ) (x) ( )
Author’s response:
“Introduction” section has been modified.
“Materials and Methods” including research design section has been modified.
“Results” section has been modified.
“Conclusions” section has been modified.
Comments and Suggestions for Authors
“The authors propose to analyze the association between the results on two neuropsychological tests (reaction time and paired-associated learning) with alcohol use in a large sample of bipolar patients (N~1200) recruited from a larger genetic study.
In their introduction, the authors expose in ONE sentence the complex issue of cognitive impairment in the course of bipolar disorder: "Cognitive impairment is a prominent [1] and generally stable [2] core symptom [3] in persons with bipolar disorder (BD)".
Thus, they miss a central caveat in assessing such neuropsychological impairment. GENERALLY STABLE means that cognitive impairment is persistent BETWEEN mood episodes. the teams that have properly measured such impairment are cautious enough to assess them in euthymic patients. for an example, see Roux et al. 2020 PMID 29734950. in that study, Roux et al. used cut-off scores of depression and mania to select euthymic patients for their study.
Of course, during manic and depressive episodes, the symptoms impair neuropsychological results, toward impulsive rapid responses during mania, and toward slow responses in depression.
Here the methods section does not provide any kind of subject selection based on euthymia checking, not even a post-hoc analysis of a correlation between the neuropsychological results and the symptoms levels.
As such, the design does not allow to draw any conclusion from the results.
the patients with higher alcohol use, or even more with an alcohol use disorders are known to have a worse course of their bipolar disorder (more episodes, more affective liability, more suicidality, less response rate to mood stabilizers). they are expected to be more symptomatic in such a large cohort. they could display different results in RT and PAL only because they are less euthymic.”
Author’s response:
Thank you for notifying us about the importance of “phase of illness” in bipolar disorder - an important issue to be considered while measuring cognitive functions in bipolar disorder.
In the original SUPER-study, all the patients who were unable to give written informed consents had been excluded. In the present study, all the hospitalized patients were excluded. In these ways, many of the bipolar disorder patients with severe manic or depressive episode had been already excluded from our analyses.
To avoid confusion, in the revised version of our manuscript we have deleted a sentence from the method describing recruitment of hospitalized patients in the SUPER-study. The deleted sentence was:” Also, patients admitted involuntarily, and forensic psychiatric patients were included in the study subject selection to get as large a sample and phenotype range as possible.”
In the revised version however, we were able to adjust depressive symptoms in our analyses by using the five-item Mental Health Inventory-5 (MHI-5) (Cuijpers P et al., 2009, https://doi.org/10.1016/j.psychres.2008.05.012). Unfortunately, we were not able to adjust manic symptoms, as the original data did not include assessment of current manic symptoms.
In the revised version, we have modified and updated abstract, introduction, materials and methods, results, conclusions, strengths, limitations, references, and also provide additional supplementary materials.
For your convenience, giving herewith an account of the major changes made in the revised version of our manuscript.
Abstract (Original result)
“After adjustment for age, education and housing status, hazardous drinking was associated with lower median and less variable RT in females while AUD was associated with a poorer PAL test performance in terms of the total errors adjusted scores in females.”
Abstract (Modified result)
“After adjustment of age, education, housing status and depression, hazardous drinking was associated with lower median and less variable RT in females while AUD was associated with a poorer PAL test performance in terms of the total errors adjusted scores in females.”
- Introduction (Original starting):
“Cognitive impairment is a prominent [1] and generally stable [2] core symptom [3] in persons with bipolar disorder (BD).”
- Introduction (Modified starting):
“Persons suffering from bipolar disorder (BD) manifest cognitive impairment specially during manic or depressive phase of the disorder, and also less intensely during euthymic phase [1,2,3].
- Materials and Methods:
2.2. Bipolar Disorder Diagnoses (A new paragraph has been added at the end)
“One of the inclusion criteria for the selection of SUPER-study population was that they should be able to sign the written informed consent themselves. Thus, many BD patients with severe manic or depressive episode were excluded. They were further excluded in the present study by including only participants living independently thus excluding those who had been hospitalized.”
2.6. Confounding Factors (Original):
“Age, education [44] and housing status [45] have effects on cognition. However, we considered age and education to be the confounding variables in this study.”
2.6. Confounding Factors (Modified):
“Age, education [44], housing status [45] and depressive symptoms [3] have effects on cognition. Hence, we considered them to be the confounding variables in this study.”
2.6.4. Depression (New confounding factor added):
“Depressive symptoms might be associated with poorer cognitive performance [3], hence we considered depression as a confounder. We used the five-item Mental Health Inventory-5 (MHI-5) to detect depressive symptoms. In the analysis MHI-5 was dichotomized. We used ≤72 cutoff score for depression which was also used in a recent population-based study in Finland [52].”
3.1. Results:
3.1. Background Factors and Alcohol Use Patterns in Persons with Bipolar Disorder (A new line has been added at the end):
“Three-fourth males and females were detected screening-positive for depression.”
Table 1. Background factors and alcohol use patterns in persons with bipolar disorder (Modified with addition of new data on depression):
|
Male |
Female |
|
|
N = 466 |
N = 793 |
|
|
Age (mean (SD)) |
45.35 (13.05) |
44.37 (12.74) |
|
Education |
|
|
|
No matriculation examination (%) |
309 (66.3) |
436 (55.0) |
|
Matriculation examination (%) |
157 (33.7) |
357 (45.0) |
|
Household pattern |
|
|
|
With spouse |
172 (36.9) |
341 (43.0) |
|
Without spouse |
294 (63.1) |
452 (57.0) |
|
DepressionΩ |
|
|
|
Depressed |
331 (71.0) |
583 (73.5) |
|
Non-depressed |
135 (29.0) |
210 (26.5) |
|
Current Psychotrophic medications |
|
|
|
No (%) |
27 (5.8) |
36 (4.5) |
|
Yes (%) |
439 (94.2) |
756 (95.3) |
|
Missing (%) |
0 (0.0) |
2 (0.2) |
|
Hazardous drinkingψ |
|
|
|
No (%) |
287 (61.6) |
584 (73.6) |
|
Yes (%) |
179 (38.4) |
209 (26.4) |
|
Lifetime alcohol-related disorder |
|
|
|
No (%) |
288 (61.8) |
597 (75.3) |
|
Yes (%)
|
178 (38.2) |
196 (24.7) |
|
SD = Standard deviation ΩMHI-5 cutoff score for depression was ≤72 |
||
3.2. Association of Reaction Time and Visual Memory with Hazardous Drinking in Bipolar Disorder (A new paragraph has been added at the end):
“The Association of reaction time and visual memory with hazardous drinking without adjusting for depression has been shown in the Supplementary Materials, Table S7. The results were basically same without depression as a covariate.”
3.3. Association of Reaction Time and Visual Memory with Alcohol Related Disorder in Bipolar Disorder (A new paragraph has been added at the end):
“The Association of reaction time and visual memory with Alcohol Related Disorder without adjusting depression has been shown in the Supplementary Materials, Table S8. The results were basically same without depression as a covariate.”
4.3. Strengths (A new paragraph has been added before the last paragraph):
“Our inclusion criterion of independent living excluded hospitalized patients, so patients with severe manic or depressive episode were not included. Another inclusion criterion was the ability to give written informed consent which also restricted inclusion of bipolar disorder patients with severe manic or depressive symptoms.”
4.4. Limitations (A new paragraph has been added before the last paragraph):
“The phase of the illness in bipolar disorder is an important issue to be considered while measuring cognitive functions in bipolar disorder. But unfortunately, we were not able to use manic symptoms in our analysis, as the data did not include assessment of current manic symptoms.”
- Conclusions (First line has been modified):
“Hazardous drinking was not associated with a cognitive decline in persons with bipolar disorder adjusted with age, education, household pattern and depression.”
Date of submission: 15 June 2021
Date of review: 19 Jul 2021 11:14:30
Date of submitting author’s response: 13 Aug 2021

Round 2
Reviewer 2 Report
The authors have taken into account some of my remarks and requests. Still, they cannot change the initial study design that was not intended to assess bipolar patients during an euthymic phase.
The article can be published provided that they are extremly cautious in the interpretation of the results. Some remaining sentence still have to be modified.
Abstract:
"Alcohol use was assessed through hazardous drinking and alcohol related disorder including alcohol use disorder (AUD) drinking was screened with the AUDIT-C (Alcohol Use Disorders Identification Test for Consumption) screening tool. Alcohol related disorder diagnoses were obtained from the national registrar
data."
here the authors should add: depressive symptoms were assessed with the MHI5. As soon as the abstract, the reader should find a sentence stating that "no assessement of current manic symptoms was available"
With all the difficulties in interpreting the results due to the methodological flaw of not assessing manic symptoms, the last sentence of the abstract: "Our findings of positive associations between alcohol use and cognition in persons with bipolar disorder are unique" is inapropriate. The authors should be cautious in the interpretation of their results and this sentence should be deleted.
Methods:
The added sentence helps clarify that the most severe patients with delusional depressive or manic episodes, hospitalized at the time of the study, could not be included. Still, the chosen sentence is particularly bold and the argument is specious: "Thus, many BD patients with severe manic or depressive episode were excluded." really? how many? do you have a flow-chart of the excluded patients? And 2/3 of the sample still have significant (above the screening) depressive symptoms. the impaiment due to BP disorder in the general population is not limited to the accute hospitalized épisodes where patients are not able to give an informed consent. Please rephrase.
Please provide the initial references for the Cambridge Neuropsychological Test Automated Battery (CANTAB) for schizophrenia. You could also provide here or in the discussion section the median RT and the PAL median FTMS of the general population study that you cite [43].
Results:
"Hazardous drinking and AUD were more common in males than in females". the reader would like to know in that section if hazardous drinking and AUD are associated with the statuts "depressed/ not depressed" in you sample defined by the MHI-5 score.
"There was no significant difference in reaction times between male and female bipolar disorder patients with or without a lifetime history of alcohol use disorder". the reader would like to know if the scores were different from the ones published in the general population.
Discussion:
"Our findings did not support our assumption that problematic drinking might be associated with impaired cognitive function in persons with bipolar disorder." Please be more cautious. You should specify "in a sample of outpatients with bipolar disorders below 70 years-old.
"On the contrary, some positive associations were found between hazardous drinking and reaction time scores in males and females. As our study is the first of its kind, it is difficult compare it to literature to compare with. However, our findings were in line with those in normal population studies [59,60]."
Does it means that patients with AUD have less depressive symptoms and are less slow that patients without AUD and more depressive symptoms? do you mean taht they are in the normal range of RT with this test?
the authors spend a lot of the discussion to review the literature on non-hazardous drinking and better results on cognitive testing, which has not much in common with the scientific question that they investigate: congitive perfromance in BP disorder patients with hazardous drinking or AUD. (P12 from "General population [REF 72-75]... to high funcitoninig non-drinkers [90]". this paragraph should be deleted.
in the limitation, the authors do not mention more than "We used only two tests from CANTAB."A more specific limitaiton acknowledgement would have mentioned that they used only a RT and a memory test, while most of the literature that they reviewed showed that AUD in BP patients is associated with EXECUTIVE FUNTIONS DEFICITS; so that they should discuss that patients with AUD may be more impulsive, and less accurate.
furthermore memory impairment is expected to occur only in patients with severe alcohol-related cognitive impairment (formerly known as alcohol dementia), a sub-group of patients with AUD.
"What does the study add?": "Hazardous drinking was not associated with cognitive decline in persons with bipolar disorder" the author should specify in outpatients, below 70 , when a memory test and a reaction time test were used whitout an assessment of executive functionning
conclusion:
" Hazardous drinking was not associated with a cognitive decline in persons with bipolar disorder adjusted with age, education, household pattern and depression.."
the author should specify in outpatients, below 70 , when a memory test and a reaction time test were used whitout an assessment of executive functionning and without correcting for manic symptoms.
"some positive associations were found between hazardous drinking and cognition, which are unique to these psychiatric disorders but in line with the findings from general population studies." this is not correct, most of the positive association are lost when adjusting (Table S6-S8), and the positive association is only in line with omderate drinking in the general population , not hazardous drinking and not AUD.
the authors should find a more cautious statement.
Author Response
Author’s responses to the comments of REVIEWER 2 second round:
English language and style
( ) Extensive editing of English language and style required
( ) Moderate English changes required
(x) English language and style are fine/minor spell check required
( ) I don't feel qualified to judge about the English language and style
Author’s response:
English language, style and spell check done accordingly.
Yes Can be improved Must be improved Not applicable
Does the introduction provide sufficient background and include all relevant references?
(X) ( ) ( ) ( )
Is the research design appropriate?
( ) (X) ( ) ( )
Are the methods adequately described?
(X) ( ) ( ) ( )
Are the results clearly presented?
( ) (X) ( ) ( )
Are the conclusions supported by the results?
( ) ( ) (x) ( )
Author’s response:
“Introduction” has been modified.
“Materials and Methods” including research design has been modified.
“Results” has been modified.
“Discussion” has been modified.
“Conclusions” has been modified.
Comments and Suggestions for Authors
The authors have taken into account some of my remarks and requests. Still, they cannot change the initial study design that was not intended to assess bipolar patients during a euthymic phase.
The article can be published provided that they are extremely cautious in the interpretation of the results. Some remaining sentence still have to be modified.
Abstract:
"Alcohol use was assessed through hazardous drinking and alcohol related disorder including alcohol use disorder (AUD) drinking was screened with the AUDIT-C (Alcohol Use Disorders Identification Test for Consumption) screening tool. Alcohol related disorder diagnoses were obtained from the national registrar data."
here the authors should add: depressive symptoms were assessed with the MHI5. As soon as the abstract, the reader should find a sentence stating that "no assessment of current manic symptoms was available"
With all the difficulties in interpreting the results due to the methodological flaw of not assessing manic symptoms, the last sentence of the abstract: "Our findings of positive associations between alcohol use and cognition in persons with bipolar disorder are unique" is inappropriate. The authors should be cautious in the interpretation of their results and this sentence should be deleted.
Methods:
The added sentence helps clarify that the most severe patients with delusional depressive or manic episodes, hospitalized at the time of the study, could not be included. Still, the chosen sentence is particularly bold, and the argument is specious: "Thus, many BD patients with severe manic or depressive episode were excluded." really? how many? do you have a flow-chart of the excluded patients? And 2/3 of the sample still have significant (above the screening) depressive symptoms. the impairment due to BP disorder in the general population is not limited to the acute hospitalized episodes where patients are not able to give an informed consent. Please rephrase.
Please provide the initial references for the Cambridge Neuropsychological Test Automated Battery (CANTAB) for schizophrenia. You could also provide here or in the discussion section the median RT and the PAL median FTMS of the general population study that you cite [43].
Results:
"Hazardous drinking and AUD were more common in males than in females". the reader would like to know in that section if hazardous drinking and AUD are associated with the status "depressed/ not depressed" in you sample defined by the MHI-5 score.
"There was no significant difference in reaction times between male and female bipolar disorder patients with or without a lifetime history of alcohol use disorder". the reader would like to know if the scores were different from the ones published in the general population.
Discussion:
"Our findings did not support our assumption that problematic drinking might be associated with impaired cognitive function in persons with bipolar disorder." Please be more cautious. You should specify "in a sample of outpatients with bipolar disorders below 70 years-old.
"On the contrary, some positive associations were found between hazardous drinking and reaction time scores in males and females. As our study is the first of its kind, it is difficult compare it to literature to compare with. However, our findings were in line with those in normal population studies [59,60]."
Does it mean that patients with AUD have fewer depressive symptoms and are less slow that patients without AUD and more depressive symptoms? do you mean that they are in the normal range of RT with this test?
the authors spend a lot of the discussion to review the literature on non-hazardous drinking and better results on cognitive testing, which has not much in common with the scientific question that they investigate: cognitive performance in BP disorder patients with hazardous drinking or AUD. (P12 from "General population [REF 72-75] ... to high functioning non-drinkers [90]". This paragraph should be deleted.
in the limitation, the authors do not mention more than "We used only two tests from CANTAB."A more specific limitation acknowledgement would have mentioned that they used only a RT and a memory test, while most of the literature that they reviewed showed that AUD in BP patients is associated with EXECUTIVE FUNTIONS DEFICITS; so that they should discuss that patients with AUD may be more impulsive, and less accurate.
Furthermore, memory impairment is expected to occur only in patients with severe alcohol-related cognitive impairment (formerly known as alcohol dementia), a sub-group of patients with AUD.
"What does the study add?": "Hazardous drinking was not associated with cognitive decline in persons with bipolar disorder" the author should specify in outpatients, below 70, when a memory test and a reaction time test were used without an assessment of executive functioning
conclusion:
"Hazardous drinking was not associated with a cognitive decline in persons with bipolar disorder adjusted with age, education, household pattern and depression."
the author should specify in outpatients, below 70, when a memory test and a reaction time test were used without an assessment of executive functioning and without correcting for manic symptoms.
"some positive associations were found between hazardous drinking and cognition, which are unique to these psychiatric disorders but in line with the findings from general population studies." this is not correct, most of the positive association are lost when adjusting (Table S6-S8), and the positive association is only in line with moderate drinking in the general population, not hazardous drinking and not AUD.
the authors should find a more cautious statement.
Author’s response:
Thank you for your recommendation for us to be extremely cautious in the interpretation of the results. Thank you also for your understanding of our limitations to assess bipolar patients during euthymic phase.
As per your recommendations, we have modified abstract, materials and methods, results, discussion, conclusions, and references in the latest revised version of our manuscript. We have deleted some confusing terms like “In contrast” in 4.1. Main Findings.
Additionally, we could not find any comparable literature studying reaction time in alcohol-related or alcohol use disorder in the general population to compare our results in 3.3. Association of Reaction Time and Visual Memory with Alcohol Related Disorder in Bipolar Disorder.
For your convenience, giving below an account of the changes done in this latest revised version.
Abstract:
Reviewer’s comments:
““Alcohol use was assessed through hazardous drinking and alcohol related disorder including alcohol use disorder (AUD) drinking was screened with the AUDIT-C (Alcohol Use Disorders Identification Test for Consumption) screening tool. Alcohol related disorder diagnoses were obtained from the national registrar data."
here the authors should add: depressive symptoms were assessed with the MHI5. As soon as the abstract, the reader should find a sentence stating that "no assessment of current manic symptoms was available"
With all the difficulties in interpreting the results due to the methodological flaw of not assessing manic symptoms, the last sentence of the abstract: "Our findings of positive associations between alcohol use and cognition in persons with bipolar disorder are unique" is inappropriate. The authors should be cautious in the interpretation of their results and this sentence should be deleted.”
Author’s response:
Abstract (Two new sentences have been added at the end of methods)
“Alcohol use was assessed through hazardous drinking and alcohol related disorder including alcohol use disorder (AUD) drinking was screened with the AUDIT-C (Alcohol Use Disorders Identification Test for Consumption) screening tool. Alcohol related disorder diagnoses were obtained from the national registrar data. Participants performed two computerized tasks from the Cambridge automated neuropsychological test battery (CANTAB) on tablet computer: the 5-choice serial reaction time task, or reaction time (RT) test and the Paired Associative Learning (PAL) test. Depressive symptoms were assessed with the MHI-5 (Mental Health Inventory with 5 items). However, no assessment of current manic symptoms was available.”
Abstract (Original conclusions)
“Our findings of positive associations between alcohol use and cognition in persons with bipolar disorder are unique.”
Abstract (Modified conclusions)
“Our findings of positive associations between alcohol use and cognition in persons with bipolar disorder are difficult to explain because of the methodological flaw of not being able to separately assess only participants in euthymic phase.”
Methods:
Reviewer’s comments:
“The added sentence helps clarify that the most severe patients with delusional depressive or manic episodes, hospitalized at the time of the study, could not be included. Still, the chosen sentence is particularly bold, and the argument is specious: "Thus, many BD patients with severe manic or depressive episode were excluded." really? how many? do you have a flow-chart of the excluded patients? And 2/3 of the sample still have significant (above the screening) depressive symptoms. the impairment due to BP disorder in the general population is not limited to the acute hospitalized episodes where patients are not able to give an informed consent. Please rephrase.
Please provide the initial references for the Cambridge Neuropsychological Test Automated Battery (CANTAB) for schizophrenia. You could also provide here or in the discussion section the median RT and the PAL median FTMS of the general population study that you cite [43].”
Author’s response:
2.2. Bipolar Disorder Diagnoses (Original last paragraph)
“One of the inclusion criteria for the selection of SUPER-study population was that they should be able to sign the written informed consent themselves. Thus, many BD patients with severe manic or depressive episode were excluded. They were further excluded in the present study by including only participants living independently thus excluding those who had been hospitalized.”
2.2. Bipolar Disorder Diagnoses (Modified last paragraph)
“While selecting the original SUPER-study population, those who were not able to sign the written informed consent themselves had been excluded. In the present study, those who were hospitalized had been excluded as well by including only participants living independently. Thus, the most severe bipolar patients with severe depressive, or manic episodes were presumed to be excluded. However, cognitive impairment due to bipolar disorder in general population is not limited to the acute hospitalized episodes where patients might be unable to give an informed consent.”
2.5. Cognitive Measures (The wrong terms “for schizophrenia” have been deleted and the reference [43] has been repositioned in the first paragraph)
“Processing speed and visual learning are the two cognitive domains affected invariably in psychotic illnesses; hence, we selected the Five-Choice Serial Reaction Time Task (5-CRTT) and the Paired Associative Learning (PAL) task from the Cambridge Neuropsychological Test Automated Battery (CANTAB) for schizophrenia for the assessment of reaction time (RT) and visual memory, respectively [43].”
Results:
Reviewer’s comments:
“"Hazardous drinking and AUD were more common in males than in females". the reader would like to know in that section if hazardous drinking and AUD are associated with the status "depressed/ not depressed" in you sample defined by the MHI-5 score.
"There was no significant difference in reaction times between male and female bipolar disorder patients with or without a lifetime history of alcohol use disorder". the reader would like to know if the scores were different from the ones published in the general population.”
Author’s response:
3.1. Background Factors and Alcohol Use Patterns in Persons with Bipolar Disorder (Two new sentences have been added after the first sentence of the second paragraph)
“Hazardous drinking and AUD were more common in males than in females. Hazardous drinking was significantly more common in depressed (MHI-5 < 70) females than in non-depressed (MHI-5 < 70) females (28.8 % vs 19.5%, p = 0.011). There was no significant difference in the prevalence of AUD between depressed and non-depresses females. In males these comparisons were statistically non-significant.”
3.3. Association of Reaction Time and Visual Memory with Alcohol Related Disorder in Bipolar Disorder (First sentence has been changed)
“There was no significant difference in reaction times between male and female bipolar disorder patients with or without a lifetime history of alcohol use disorder. Median RT or SD RT did not differ statistically significantly between participants with or without a lifetime history of alcohol use disorder, in males or in females (Table 3).”
Discussion:
Reviewer’s comments:
“""Our findings did not support our assumption that problematic drinking might be associated with impaired cognitive function in persons with bipolar disorder." Please be more cautious. You should specify "in a sample of outpatients with bipolar disorders below 70 years-old.
"On the contrary, some positive associations were found between hazardous drinking and reaction time scores in males and females. As our study is the first of its kind, it is difficult compare it to literature to compare with. However, our findings were in line with those in normal population studies [59,60]."
Does it mean that patients with AUD have fewer depressive symptoms and are less slow that patients without AUD and more depressive symptoms? do you mean that they are in the normal range of RT with this test?
the authors spend a lot of the discussion to review the literature on non-hazardous drinking and better results on cognitive testing, which has not much in common with the scientific question that they investigate: cognitive performance in BP disorder patients with hazardous drinking or AUD. (P12 from "General population [REF 72-75] ... to high functioning non-drinkers [90]". This paragraph should be deleted.
in the limitation, the authors do not mention more than "We used only two tests from CANTAB."A more specific limitation acknowledgement would have mentioned that they used only a RT and a memory test, while most of the literature that they reviewed showed that AUD in BP patients is associated with EXECUTIVE FUNTIONS DEFICITS; so that they should discuss that patients with AUD may be more impulsive, and less accurate.
Furthermore, memory impairment is expected to occur only in patients with severe alcohol-related cognitive impairment (formerly known as alcohol dementia), a sub-group of patients with AUD.
"What does the study add?": "Hazardous drinking was not associated with cognitive decline in persons with bipolar disorder" the author should specify in outpatients, below 70, when a memory test and a reaction time test were used without an assessment of executive functioning.”
Author’s response:
4.1. Main Findings (Original first paragraph)
“Our findings did not support our assumption that problematic drinking might be associated with impaired cognitive function in persons with bipolar disorder. On the contrary, some positive associations were found between hazardous drinking and reaction time scores in males and females. As our study is the first of its kind, it is difficult compare it to literature to compare with. However, our findings were somewhat comparable with the findings of some normal population studies suggesting that long-term moderate alcohol consumption was not associated with cognitive impairment [59,60]”
4.1. Main Findings (Modified first paragraph)
“Our findings did not support our assumption that problematic drinking might be associated with impaired cognitive function in a sample of outpatients with bipolar disorders below 70 years old. On the contrary, some positive associations were found between hazardous drinking and reaction time scores in males and females. As our study is new of its kind, it is difficult compare it to literature to compare with. However, our findings were partly in line with the findings of some general population studies suggesting that moderate drinking was not associated with cognitive impairment [59,60], not hazardous drinking and not AUD.”
4.2. Comparison with Other Studies (Two paragraphs reviewing literature on non-hazardous drinking and better results on cognitive testing have been deleted)
“General population cross-sectional studies investigating effects of alcohol on cognitive function revealed that moderate to heavy drinking was associated with cognitive de-cline [72,73,74,75] and light to moderate drinking was associated with either no effects on cognition [59,76] or cognitive enhancement [60,75,77,78,79]. Light, moderate, and heavy drinking can be defined operationally as 1.2, 2.2, 3.5 drinks/day respectively [80]. However, these definitions vary considerably worldwide.
Most cohort studies in the general population addressing the same issue revealed a positive correlation between light / light to moderate alcohol use and cognitive function [32,33,81,82,83] whereas other cohort study found no association between light to moderate alcohol consumption and better or worse cognitive functions [84,85,86]. One study re-ported a positive association between moderate to heavy drinking and cognitive function [87]. In contrast, another cohort study revealed negative association between heavy alcohol use and cognitive function in normal population [88]. One cohort study found dose-response positive association of alcohol use and cognitive function compared to abstainers and former drinkers [89]. Another cohort study revealed significant cognitive impairment in low functioning non-drinkers and light to moderate drinkers and high functioning non-drinkers [90].”
4.4. Limitations (Original first paragraph)
“Our study was cross-sectional, not longitudinal. We used only two tests from CANTAB. We did not adopt more comprehensive approach to measure working memory performance.”
4.4. Limitations (Modified first paragraph)
“Our study was cross-sectional, not longitudinal. We did not adopt more comprehensive approach to measure working memory performance. We used only a reaction time test and a memory test from the CANTAB, while most of the literature we reviewed showed that AUD in BD patients was associated with executive function deficits. So, patients with AUD might be more impulsive, and less accurate. Furthermore, memory impairment is expected to occur only in patients with severe alcohol-related cognitive impairment (formerly known as alcohol dementia), a sub-group of patients with AUD.”
4.6. What Does This Study Add? (Original)
“Hazardous drinking was not associated with cognitive decline in persons with bipolar disorder.”
4.6. What Does This Study Add? (Modified)
“Hazardous drinking was not associated with cognitive decline in persons with bipolar disorder in outpatients, below 70, when a reaction time test and a memory test were used without an assessment of executive functioning and without correcting for manic symptoms.”
Conclusion:
Reviewer’s comments:
“"Hazardous drinking was not associated with a cognitive decline in persons with bipolar disorder adjusted with age, education, household pattern and depression."
the author should specify in outpatients, below 70, when a memory test and a reaction time test were used without an assessment of executive functioning and without correcting for manic symptoms.
"some positive associations were found between hazardous drinking and cognition, which are unique to these psychiatric disorders but in line with the findings from general population studies." this is not correct, most of the positive association are lost when adjusting (Table S6-S8), and the positive association is only in line with moderate drinking in the general population, not hazardous drinking and not AUD.
the authors should find a more cautious statement.”
Author’s response:
- Conclusions (Original first paragraph)
“Hazardous drinking was not associated with a cognitive decline in persons with bipolar disorder adjusted with age, education, household pattern and depression. Rather, some positive associations were found between hazardous drinking and cognition, which are unique to these psychiatric disorders but in line with the findings from general population studies.”
- Conclusions (Modified first paragraph)
“Hazardous drinking was not associated with a cognitive decline in persons with bipolar disorder in outpatients, below 70, adjusted with age, education, household pattern and depression, when a reaction time test and a memory test were used without an assessment of executive functioning and without correcting for manic symptoms. Although, some positive associations were found between hazardous drinking and cognition, most of them were lost when adjusting (Table S6-S8), and the positive association was only in line with moderate drinking in the general population, not hazardous drinking and not AUD.”
Date of submission: 15 June 2021
Date of review: 19 Aug 2021 18:59:25
Date of submission of author’s response: 23 Aug 2021
